# Near infrared fluorescent peptide nanoparticles for enhancing esophageal cancer therapeutic efficacy

Zhen Fan[1,2,3,4,5], Yan Chang[6], Chaochu Cui[6,7,8], Leming Sun[9], David H. Wang[10,11], Zui Pan [6] & Mingjun Zhang[3,4,12]

Various types of nanoparticles have been proposed for targeted drug delivering, imaging, and tracking of therapeutic agents. However, highly biocompatible nanoparticles with structure-induced fluorescence and capability to conjugate with biomarkers and drugs remain lacking. This research proposes and synthesizes fluorescent nanoparticles (f-PNPs) assembled by cyclic peptides to combine imaging and drug delivering for esophageal cancer (EC). To achieve tumor targeting, f-PNPs are first conjugated with RGD moieties to selectively target EC cells via $\alpha_v\beta_3$ integrin; the nanoparticles are then embedded with epirubicin (EPI). Cell viability assays and analysis of tissue histology reveal that EPI-loaded RGD-f-PNPs (RGD-f-PNPs/EPI) led to significantly reduced cardiotoxicity and improved anti-tumor activity compared to EPI alone. Moreover, the drug delivery to tumor sites and therapeutic responses could be monitored with near-infrared fluorescence using RGD-f-PNPs/EPI. This unique nanoparticle system may lead to potential approaches for bioorganic fluorescence-based delivering, imaging, and drug release tracking.

[1] Department of Polymeric Materials, School of Materials Science and Engineering, Tongji University, Shanghai 201804, China. [2] Institute for Advanced Study, Tongji University, Shanghai 200092, China. [3] Department of Biomedical Engineering, College of Engineering, The Ohio State University, Columbus, OH 43210, USA. [4] Dorothy M. Davis Heart and Lung Research Institute, The Ohio State University Wexner Medical Center , OH, 43210 Columbus, Ohio, USA. [5] Comprehensive Cancer Center, The Ohio State University, 43210 Columbus, OH, USA. [6] College of Nursing and Health Innovation, The University of Texas at Arlington, Arlington, TX 76019, USA. [7] State Key Laboratory of Oncology in South China, Collaborative Innovation Center for Cancer Medicine, Sun Yat-sen University Cancer Center, 510060 Guangzhou, China. [8] Henan Key Laboratory of Medical Tissue Regeneration, Xinxiang Medical University, 453003 Xinxiang, Henan, China. [9] School of Life Sciences, Northwestern Polytechnical University, 710065 Xi'an, China. [10] Department of Internal Medicine, Esophageal Diseases Center, University of Texas Southwestern Medical Center, Dallas, TX 75390, USA. [11] VA North Texas Health Care System, Dallas, TX 75216, USA. [12] Interdisciplinary Biophysics Graduate Program, The Ohio State University, Columbus, OH 43210, USA. These authors contributed equally: Zhen Fan, Yan Chang, Chaochu Cui. Correspondence and requests for materials should be addressed to Z.P. (email: zui.pan@uta.edu) or to M.Z. (email: zhang.4882@osu.edu)

Esophageal cancer (EC) is the sixth leading cause of cancer mortality worldwide[1]. In 2018, 15,850 deaths from EC is estimated for the United States alone[2]. There are two main types of EC: adenocarcinoma (EAC) and squamous cell carcinoma (ESCC). While EAC is more common in the United States and Western Europe, ESCC predominates globally with a higher incidence reported in Asia and developing countries[3, 4]. Since there are minimal symptoms during the early stages of EC, most patients are diagnosed at late stages which limit curative treatment options[5]. Chemotherapy is often used to slow the growth of tumor and relieve cancer symptoms. Whereas platinum drugs and taxanes are routinely used chemotherapeutic agents for EC, the anthracycline compound, epirubicin (EPI), is sometimes administered to EC patients with a good performance status. Due to lack of cell specificity, many chemotherapeutic drugs have significant toxicities limiting dosing frequency and cumulative lifetime dose. For example, EPI can cause cardiac toxicity, bone marrow suppression, and secondary leukemia[6–9]. Therefore, there is an urgent need for effective treatments integrated with targeted drug delivery strategies to minimize drug side effects[10].

A possible strategy is to directly target chemotherapeutic compounds to tumors using nanoparticles. Nanoparticles can be enriched in tumor tissues via enhanced permission and retention effect and prolong drug half-life, improve solubility of hydrophobic drugs, and reduce potential immunogenicity[11]. Many nanoparticle-based imaging systems require complex designs and engineering of external fluorescence imaging agents, for example, organic fluorophores or quantum dots (QDs)[12]. Since organic dyes have photobleaching issues limiting their clinical applications[13, 14], much research on fluorescent nanoparticles has been focused on QDs due to their predictable and stable fluorescence properties[15–17]. However, the use of heavy metals involved QDs also raises concerns on biocompatiblity[18, 19]. Biocompatible peptide-based nanosystems have been proposed to demonstrate the fluorescence due to the assembled nanostructure[20–22]. In most reports on fluorescent peptide-based materials, peptides are usually utilized as functional agents for their biological activities[23–26]. Meanwhile, the fluorescent cyclic peptide nanoparticles (f-PNPs) developed in our group possess fluorescence property themselves and do not require additional modification with QDs or fluorophores. As self-assembled peptide nanoparticles were made of natural amino acids, which have inherent biocompatiblity, peptide self-assemblies are biodegradable in physiological conditions and convenient for further modification or loading with therapeutic or targeting agents owing to its chemical diversity. This research group recently designed and fabricated zinc-coordinated fluorescent (peak emission wavelength: 423 nm) dipeptide nanoparticles[27]. The study suggested the fluorescent dipeptide nanoparticle as a functional nanoprobe for targeted cancer cell imaging and real-time monitoring of the drug release; however, tuning the fluorescence of the peptide nanoparticles to longer wavelength, such as near infrared (NIR) range, where light has deeper tissue penetration and opens broad opportunities for in vivo clinical applications, remains challenging. It is necessary to optimize the design of peptide building blocks and self-assembly process to obtain NIR fluorescence property for peptide nanoparticles.

In this study, visible and NIR f-PNPs, assembled by cyclo-[-(D-Ala-L-Glu-D-Ala-L-Trp)$_2$-] peptides, are designed, fabricated, and experimentally validated for drug delivery and imaging. The self-assembled f-PNPs have been characterized for their nanostructures, optical properties, efficacy, and biosafety. Furthermore, the nanoplatform is designed with two additional features to achieve EC tumor targeting for both imaging and drug delivery. First, these f-PNPs are conjugated with RGD peptide moieties to provide tumor targeting ability (Fig. 1a). These RGD-conjugated f-PNPs (RGD-f-PNPs) are further embedded with EPI via π–π stacking and electrostatic interactions between chemo-drug and peptides. As illustrated in Fig. 1b, the EPI embedded RGD-f-PNPs (RGD-f-PNPs/EPI) nanoconjugates tend to accumulate more in the tumor tissue compared to normal tissues due to the enhanced permeability and retention effects. Moreover, RGD peptide moieties bind to the overexpressed $\alpha_v\beta_3$ integrin subunits and internalize into EC cells. The embedded EPI will be released from the RGD-f-PNPs and will eventually accumulate in the nucleus to kill cancer cells. The tumor targeting and enhanced cancer cell internalization capabilities of the RGD-f-PNPs/EPI are characterized in vitro using EC cells. The targeted delivery and tracking of EPI are demonstrated in vivo using xenograft EC mice model. Through both the in vitro and in vivo experiments, an enriched anti-cancer activity and reduced systemic toxicity of EPI are achieved via localized drug delivery using RGD-f-PNPs/EPI. In addition, the drug delivery to tumor sites and therapeutic responses could be monitored in vivo with NIR fluorescence from RGD-f-PNPs/EPI. As a result, the RGD-f-PNPs could serve as a biocompatible nanoplatform for both EC imaging and enhanced therapeutic effects.

## Results

**Synthesis and characterization of the RGD-f-PNPs/EPI.** Cyclic octa-peptide cyclo-[(D-Ala-L-Glu-D-Ala-L-Trp)$_2$-] and Zn$^{2+}$ ions were co-assembled to form spherical nanoparticles under controlled experimental conditions. The goal was to create Zn$^{2+}$-coordinated cyclic peptide nanoparticles, so that fluorescence could be generated by quantum confinement within the nanostructures. The Zn$^{2+}$ ion was chosen as the metal connector due to its high affinity for complexation with carboxyl groups[28]. In the meantime, zinc coordination helps to stabilize the zinc chelation structure and limit further energy dissipation during thermal relaxation pathways for better quantum yield and fluorescence intensity[28, 29]. This was inspired by zinc coordination discovered in zinc-finger proteins, where Zn$^{2+}$ ion is coordinated at the sites by the oxygen atoms of glutamic acids[30]. Here, two glutamic acids were incorporated into the cyclic peptide as linkers to conjugate with Zn$^{2+}$ ions. As tryptophan possesses the highest amount of π electrons amongst the 20 standard amino acids, two tryptophans were chosen to provide delocalized π electrons for fluorescence. The π–π stacking interactions between the tryptophans can serve as the driving force for self-assembly and also increase electronic polarizability[31, 32]. The remaining four amino acids of cyclic octa-peptide were D-alanines, which helped to form the flat ring structure. Through further optimization of the self-assembly parameters, both images of scanning electron microscopy and atomic force microscopy confirmed the spherical nanoparticles with an average diameter of approximately 28 nm (Fig. 2a, b), which is significantly smaller than most self-assembled peptide nanostructures reported in the literature[33, 34]. As shown in Supplementary Fig. 1, the FTIR spectra of cyclo[-(D-Ala-L-Glu-D-Ala-L-Trp)$_2$-] assemblies displayed amide I bands that correlate well with those expected for the hydrogen-bonded antiparallel β-sheet structure, with peaks at 1634.8 and 1684.9 cm$^{-1}$ [35]. In addition, powder X-ray diffraction (PXRD) patterns of f-PNPs exhibited distinct sharp peaks with high intensities, indicating highly crystallized nanostructures (Fig. 2c). Overall, the FTIR and PXRD spectra indicated well-ordered nanocrystal structures with repeated nanolattices formed after tunable co-assembly of cyclic octa-peptides and Zn$^{2+}$ ions. Fluorescence vs. excitation spectra indicated both visible and NIR emission signals under various excitation wavelengths (Fig. 2d). Fluorescence emission spectra of pure water were also measured with excitation wavelength from 370 to 790 nm with intervals of

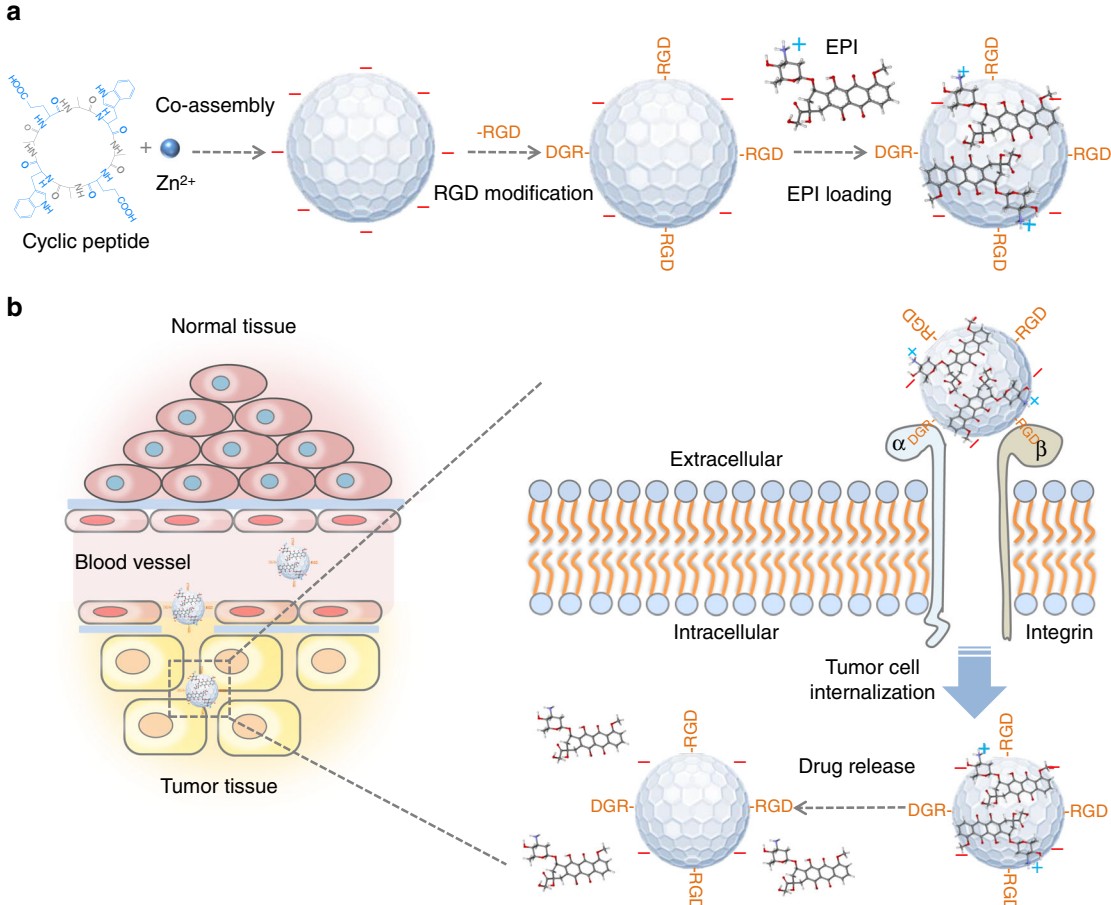

**Fig. 1** A schematic illustration of the synthesis of RGD-f-PNPs/EPI and its targeted EPI delivery into EC cells. **a** The RGD-f-PNPs were first synthesized by co-assembly $Zn^{2+}$ ions and cyclo[-(D-Ala-L-Glu-D-Ala-L-Trp)$_2$-] peptides, and then modified with RGD peptide moieties onto the surface of f-PNPs. The EPI was loaded with RGD-f-PNPs through π–π stacking and electrostatic interactions. **b** The EPI-loaded RGD-f-PNPs can be used for targeted imaging and destruction of EC cells due to their capability of actively targeting and enhanced penetration. Specifically, the RGD-f-PNPs/EPI nanoconjugates tend to accumulate in the tumor tissue compared to normal tissues due to the enhanced permission and retention (EPR effects). Moreover, RGD peptide moieties bind to the overexpressed $\alpha_v\beta_3$ integrin subunits and internalize into EC cells. The embedded EPI will be released from the RGD-f-PNPs and eventually accumulates in the nucleus to kill the cancer cells

10 nm (Supplementary Fig. 2) to exclude the fluorescence background from water. The comparative tests of the f-PNPs with the commercially used NIR organic dye indocyanine green (ICG) were conducted. Photostability evaluation of the f-PNPs indicated that the fluorescence intensity of the f-PNPs remained stable after continuous irradiation for about 300 s (Supplementary Fig. 3). On the contrary, clear fluorescence decay was observed for ICG during 300 s continuous irradiation. Different from other drug delivery systems reported in the literature[36–39], these self-assembled peptide nanoparticles were made of natural amino acids, which have inherent biocompatiblity, and likely will raise less health or environmental concern. In addition, the visible and NIR fluorescence properties of f-PNPs provided a promising in vitro and in vivo molecular imaging capacity.

RGD, an arginine-glycine-aspartic acid tri-peptide binds with high affinity to integrin $\alpha_v\beta_3$ which is often overexpressed on the tumor neovasculature and tumor cells[40]. Like other solid tumors, the tumor tissues in EC patients also present increased expression of integrin $\alpha_v\beta_3$ in both vasculature and cancer cells[41, 42]. Meanwhile, RGD has been proven to be an efficient motif to bind to the overexpressed $\alpha_v\beta_3$ integrin subunits and internalize into EC cells for enhanced tumor homing and cancer cell uptake[43, 44]. Since RGD peptides have been well demonstrated to have high binding affinity with $\alpha_v\beta_3$ integrin, we chose this peptide moiety

to enhance EC tumor targeting capability. The f-PNPs were modified by conjugation of carboxyl terminated f-PNPs and amine terminated RGD peptide under catalyst of 1-ethyl-3-(3-dimethylaminopropyl) carbodiimide and N-hydroxysuccinimide[45–47]. Mass spectrum was utilized to verify the conjugation between RGD and f-PNPs (Fig. 2e). In addition, the size distribution of the RGD-f-PNPs demonstrated that their average diameter was increased from 28.18 to 30.31 nm after RGD modification (Supplementary Fig. 4). The RGD-f-PNPs were able to emit visible and NIR fluorescence when excited with light of 370 and 760 nm (Fig. 2f, g). The above data also showed no obvious differences of the fluorescence in f-PNPs and RGD-f-PNPs. Next, RGD-f-PNP to carry a drug load of EPI was designed. Through π–π stacking and electrostatic interactions, EPI had a tendency to stack with the aromatic RGD-f-PNPs (Fig. 1a). The conjugation of the EPI with RGD-f-PNPs was characterized using the fluorescence and absorbance spectrometer (Fig. 2h, i and Supplementary Fig. 5). Clear absorbance and fluorescence quenching were observed in both visible and NIR ranges after EPI loading due to the electrostatic interactions between the EPI and RGD-f-PNPs[48]. Similarly, the fluorescence of both RGD-f-PNPs and EPI would be regained upon drug released from the nanoparticles. The observable change of optical properties of the RGD-f-PNPs with and without EPI loading

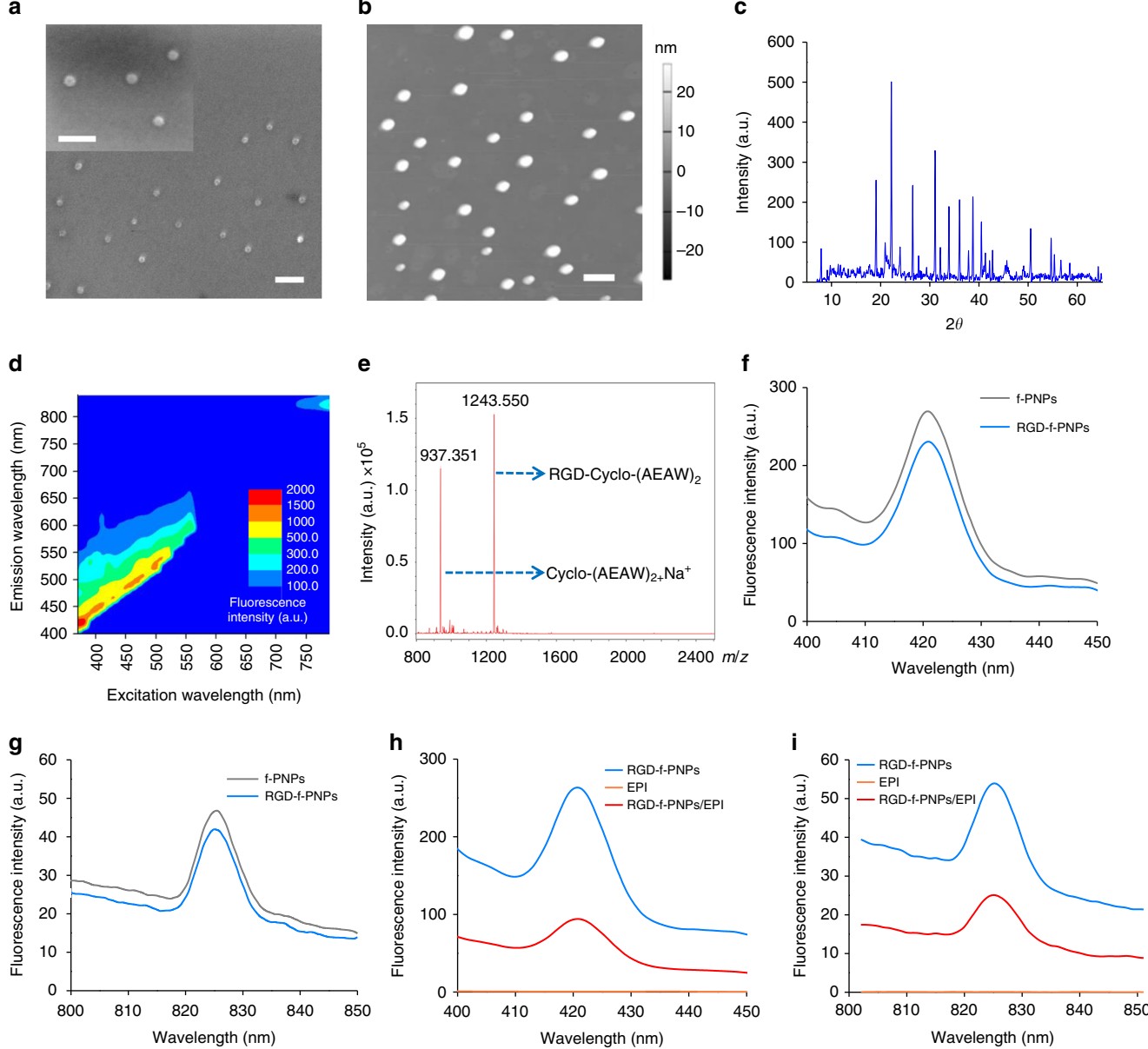

**Fig. 2** Nanomorphological and optical properties of the f-PNPs and RGD-f-PNPs/EPI complex. **a**, **b** SEM and AFM image of the f-PNPs indicate an average diameter of ~28 nm, which is significantly smaller than most self-assembled peptide nanostructures reported in the literature. Scale bar: 100 nm. **c** PXRD spectrum with distinct sharp peaks and high intensities indicates well-ordered nanocrystal structure of the f-PNPs. **d** Fluorescence vs. excitation spectra indicating multiple emission signals under various excitation wavelengths. Both visible and NIR fluorescence emissions were observed. **e** Mass spectra of RGD-f-PNPs after nanoparticle disassembly. The molecular weight of 1243.550 indicates the successful conjugation between RGD and cyclo[-(D-Ala-L-Glu-D-Ala-L-Trp)$_2$-] peptides. The molecular weight of 937.351 indicates the unconjugated cyclo[-(D-Ala-L-Glu-D-Ala-L-Trp)$_2$-] peptides. **f**, **g** Fluorescence emission spectra of the RGD-f-PNPs. Excitation wavelengths are 370 and 760 nm, respectively. No obvious fluorescence change was observed for f-PNPs after the RGD modification. **h**, **i** Fluorescence emission spectra of the EPI alone, RGD-f-PNPs before and after EPI loading. Excitation wavelengths are 370 and 760 nm, respectively. Obvious fluorescence quenching was observed after EPI loading

makes it possible to monitor the drug activities and track the drug release process in real time using fluorescence microscopy.

**Enhanced EC cell targeting and internalization using RGD-f-PNPs.** Once the RGD-f-PNPs/EPI was fabricated and characterized, human EC cell lines were used to evaluate the effects of cellular targeting and internalization of RGD-f-PNPs. The expressions of integrin $\alpha_v\beta_3$ in human normal esophageal epithelial (HET-1A), EAC (OE33), and ESCC (KYSE-30) cell lines were compared. Using specific antibodies against integrin $\alpha_v\beta_3$

(integrin $\alpha_v$ is also called CD51)[49], flow cytometry and immunofluorescent high-resolution confocal microscopy imaging were performed. As shown in Fig. 3a, the fluorescence signals from OE33 and KYSE-30 cells clearly shifted to the right compared to that from HET-1A cells, indicating that EAC and ESCC cells have higher expression levels of plasma membrane integrin $\alpha_v\beta_3$ than HET-1A cells. Similar results were obtained from immunofluorescent microscopy imaging (Supplementary Figs. 6 and 7). Compared to HET-1A cells, OE33 and KYSE-30 cells showed much higher fluorescence intensity at cell surface (Supplementary Figs. 6 and 7). In OE33 cells transfected with pX330-Cas9

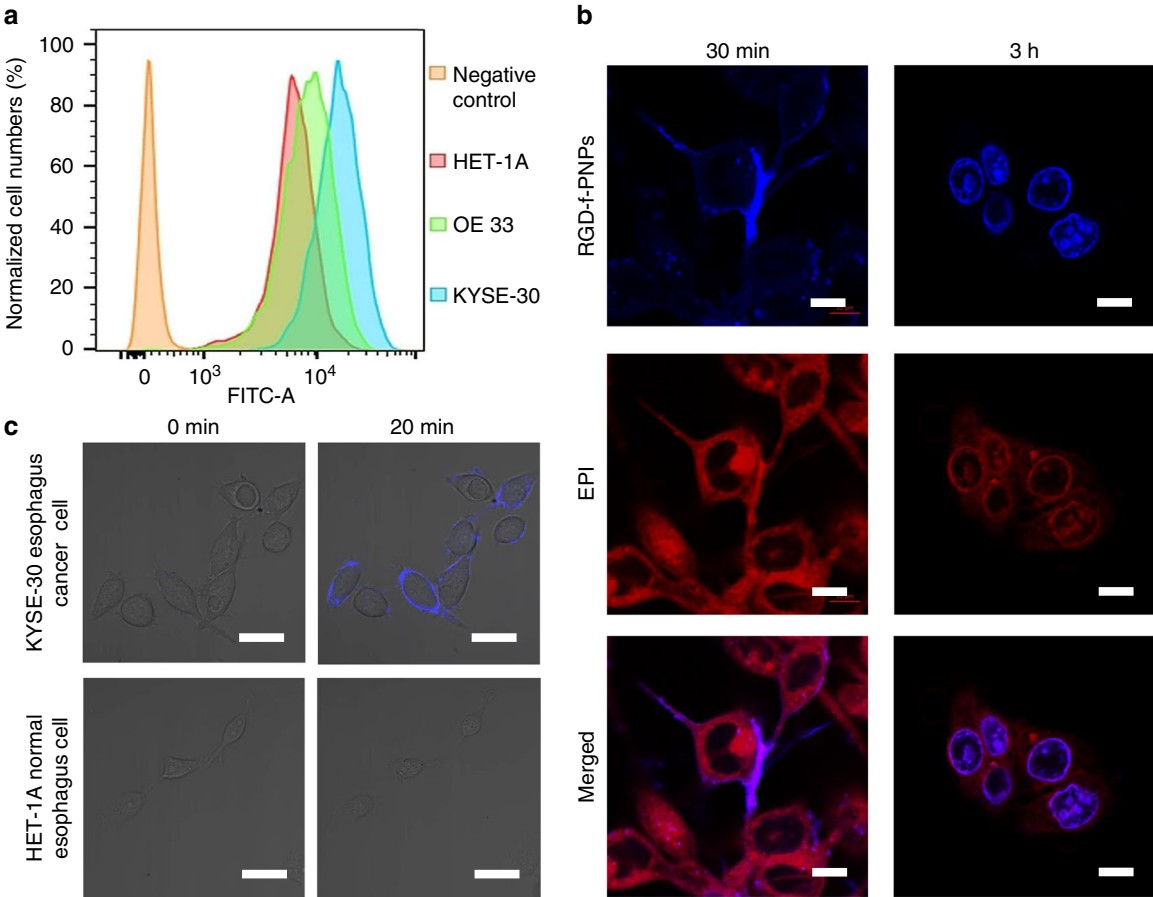

**Fig. 3** In vitro targeted EC cell imaging and enhanced internalization. **a** Flow cytometry of HET-1A, OE33, and KYSE-30 cells indicates that the $\alpha_v\beta_3$ integrins are overexpressed in EC cells. The fluorescence signals from OE33 and KYSE-30 cells shifted clearly to right compared to that from HET-1A cells, indicating that EAC and ESCC cell lines contain higher expression of plasma membrane integrin $\alpha_v\beta_3$ than HET-1A cells. **b** Confocal fluorescence images of the KYSE-30 cells after incubation with RGD-f-PNPs/EPI conjugates for 30 min and 3 h. The blue and red color represents the fluorescence signal of the RGD-f-PNPs and EPI, respectively. After 30 min of treatment, a significant blue fluorescence was observed from the KYSE-30 plasma membranes, which can be attributed to the binding between the RGD peptide moieties and overexpressed integrins at the plasma membrane. The data suggest that RGD peptide moieties could enhance the specific binding between RGD-f-PNPs and plasma membrane integrins. After 3 h of the treatment, clear red fluorescence was observed at the cell nucleus, indicating that RGD-f-PNPs/EPI has internalized into the cell nucleus. Scale bar: 10 μm. **c** Confocal fluorescence images of KYSE-30 and HET1-A cells incubated with RGD-f-PNPs after 0 and 20 min. A significant blue fluorescence of the RGD-f-PNPs was only observed from the KYSE-30 cell membrane after 20 min of treatment, which can be attributed to the specific binding between RGD moiety and integrin on the cell membrane. The data suggest that RGD-f-PNPs can selectively target EC cells vs. normal epithelial cells. Scale bar: 20 μm

plasmids containing guide RNA (gRNA)-specific targeting gene encoding CD51, the immunofluorescence signals were missing, indicating that the cell surface immunoflurescent staining was indeed from CD51 (Supplementary Fig. 7). These results suggest that both ESCC and EAC cells have significantly higher levels of integrin $\alpha_v\beta_3$ expressed at the plasma membrane compared with HET-1A cells.

Next, whether RGD-f-PNPs/EPI could selectively target EC cells was examined. The blue and NIR fluorescence of RGD-f-PNPs make it possible to monitor the cellular binding, uptake, and intracellular trafficking, as well as EPI transportation and location using confocal fluorescence microscopy. High-resolution confocal microscopy images of the KYSE-30 cells were collected after incubation with the RGD-f-PNPs/EPI. The blue and red colors represent the fluorescence signals of the RGD-f-PNPs and EPI, respectively. As shown in Fig. 3b, a strong blue fluorescence with plasma membrane pattern was observed in KYSE-30 cells 30 min after the addition of RGD-f-PNPs/EPI into the culture medium. Similar RGD-f-PNPs plasma membrane staining was also observed in OE33 cells, but not in OE33 cells transfected with pX330-Cas9 plasmids containing gRNAs targeting *CD51* gene

(Supplementary Fig. 7). After 3 h of treatment with RGD-f-PNPs/EPI, clear red fluorescence was observed within the cell nucleus, indicating that the RGD-f-PNPs/EPI had been internalized into the cells and their nuclei. Further, live cell imaging revealed that the RGD-f-PNPs/EPI could accumulate around and bound to the plasma membrane as early as 20 min when incubated with OE33 or KYSE-30 cells, but not with HET-1A cells (Fig. 3c and Supplementary Fig. 7). These data suggested that RGD peptide moieties enhance the specific binding between nanoparticles and EC cells, which facilitate selective targeting of RGD-f-PNPs/EPI to EC cells vs. normal epithelial cells.

**In vitro EPI release tracking and enhanced anti-tumor effect.** As discussed earlier, drug release can be monitored by measuring the diminished absorbance and fluorescence quenching between the RGD-f-PNPs and EPI. Confocal microscopic live cell imaging was conducted in KYSE-30 cells that were incubated with RGD-f-PNPs/EPI nanoconjugates for 5 min, 1 h, and 3 h, respectively (Fig. 4a). After 5 min, the fluorescence was almost negligible. With a prolonged incubation time, more EPI was released and,

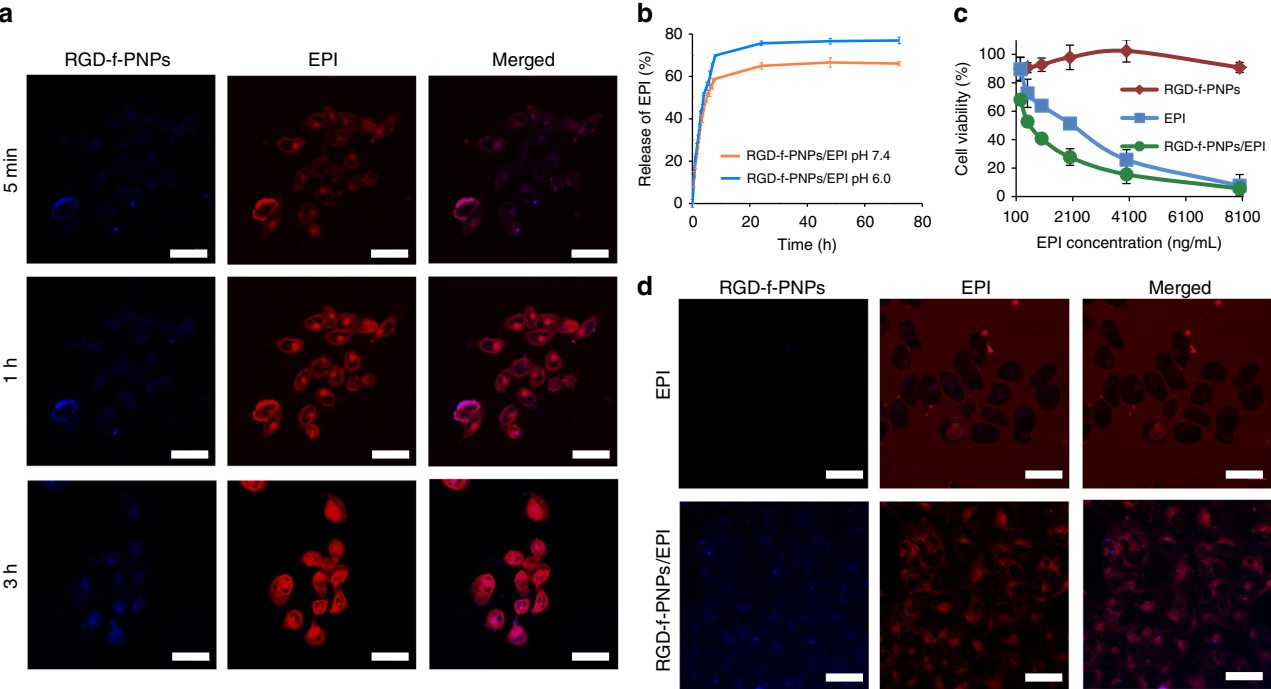

**Fig. 4** In vitro drug release monitoring and enhanced anti-tumor efficacy. **a** Confocal fluorescence images of KYSE-30 cells after being incubated with the RGD-f-PNPs/EPI conjugates for 5 min, 1 h, and 3 h. With a prolonged incubation time, more EPI was released. Consequently, the fluorescence intensity of both the RGD-f-PNPs and EPI gradually increased. The data suggest that it is possible to visualize the drug-releasing process using RGD-f-PNPs as a nanocarrier. Scale bar: 40 μm. **b** The EPI release kinetics from the RGD-f-PNPs at room temperature in PBS at pH 7.4 and 6 was measured and plotted with respect to the change of the absorbance intensity at 480 nm. The results indicated that more EPI was released in an acidic environment. Error bars represent s.d. ($n = 3$). **c** Viability of KYSE-30 cells after being treated with RGD-f-PNPs alone, EPI alone, and RGD-f-PNPs/EPI at different concentrations. Without EPI loading, the RGD-f-PNPs indicate low cytotoxicity. The cytotoxicity of RGD-f-PNPs/EPI was significantly higher than EPI alone in KYSE-30 cells, in which RGD-f-PNPs had selective targeting through RGD-integrin $\alpha_v\beta_3$ subunit binding. Especially at lower drug dose, RGD-f-PNPs/EPI indicated more significantly enhanced anti-tumor efficiency compared to EPI alone. Error bars represent s.d. ($n = 3$). **d** Confocal fluorescence images of KYSE-30 cells incubated with RGD-f-PNPs/EPI and EPI alone. A much brighter intracellular fluorescence of EPI was observed in cells incubated with RGD-f-PNPs/EPI vs. cells incubated with EPI only, indicating that the RGD-f-PNPs could facilitate the cellular uptake and intracellular accumulation of EPI into EC cells. In addition, a different internalization pattern was observed when treated with EPI alone compared to EPI-loaded RGD-f-PNPs. Scale bar: 40 μm

consequently, the fluorescence signal of both the EPI (red) and RGD-f-PNPs (blue) gradually increased. After 3 h of incubation, a relatively intense fluorescence signal was observed. Besides visualizing the drug release process, quantitating the amount of drug released is also important to regulate anti-tumor effects. In general, drug release is influenced or triggered by stimuli in the microenvironment. As tumor tissue has an acidic extracellular microenvironment as a consequence of tissue hypoxia[50, 51], larger amounts of EPI are released at lower pH values favoring the cancer killing effects. The release behavior of EPI from the RGD-f-PNPs at room temperature in phosphate-buffered saline (PBS) at pH 7.4 and 6 was measured and plotted with respect to the change of the absorbance intensity at 480 nm (Fig. 4b). The results showed that more EPI was released in an acidic environment (e.g., extracellular matrix in tumor tissues), which has the potential to enhance anti-tumor efficacy.

Next, the viabilities of KYSE-30 cells treated with RGD-f-PNPs alone, EPI alone, and RGD-f-PNPs/EPI at different concentrations were compared by using the 3-(4,5-dimethylthiazol-2-yl)-2,5-diphenyltetrazolium bromide (MTT) assay. While RGD-f-PNPs alone had negligible impact on cell viability (upper curve), both EPI and RGD-f-PNPs/EPI showed significant dose-dependent cell killing effects on the KYSE-30 cells (Fig. 4c). It is worthwhile to mention that the cell killing effects of RGD-f-PNPs/EPI was significantly higher than EPI alone in KYSE-30 cells, in which RGD-f-PNPs had selective targeting through RGD-integrin $\alpha_v\beta_3$ subunit binding. Especially at lower drug

dose, RGD-f-PNPs/EPI indicated more significantly enhanced anti-tumor efficiency compared to the EPI alone. To further visualize cellular EPI delivery, KYSE-30 cells were incubated with RGD-f-PNPs/EPI and EPI, respectively. Confocal microscopic imaging was conducted 15 min after incubation with both RGD-f-PNPs/EPI and with EPI alone. As shown in Fig. 4d, a much brighter intracellular fluorescence of EPI was observed in cells incubated with RGD-f-PNPs/EPI vs. cells incubated with EPI only, indicating that the RGD-f-PNPs could facilitate the cellular uptake and intracellular accumulation of EPI into the EC cells. In addition, a different internalization pattern was observed when treated with EPI alone compared to EPI-loaded RGD-f-PNPs.

**In vivo enhanced anti-tumor efficacy of RGD-f-PNPs/EPI.** Tumor-bearing nude mice were tail vein injected with RGD-f-PNPs, RGD-f-PNPs/EPI, or EPI to study their delivery efficacy. The tumors were produced by injecting $1 \times 10^6$ KYSE-30 cells per animal into 6-week-old nude mice. When the tumors reached a diameter of about 5 mm, the tumor-bearing mice were randomly divided into four treatment groups: control (saline only), EPI, RGD-f-PNPs, and RGD-f-PNPs/EPI. Interestingly, no tumor was observed in mice injected with KYSE-30 cells lacking CD51 within 4 weeks. It suggests that CD51 is essential for EC tumor formation. To verify the enhanced anti-tumor effects of EPI combined with RGD-f-PNPs, mice were treated with various drug formations at the dose of 1.5 mg kg$^{-1}$ RGD-f-PNPs/EPI and 6

mg kg$^{-1}$ EPI for 3 weeks. No obvious side effects for the RGD-f-PNPs were observed in terms of signs of distress, lethality, or significant drop in body weight, while treatment of EPI at the indicated dose (6 mg kg$^{-1}$) resulted in significant loss in body weight and three out of five animals died after 1 week of treatment and the remaining two animals reached poor body condition scores for euthanasia (Fig. 5a). Hematoxylin and eosin (H&E) staining was performed on postmortem tissues to examine the EPI-induced damage in the hearts, livers, kidneys, spleens, and esophagi (Supplementary Fig. 8). Normal morphology was observed in the spleens and esophagi from control, EPI, and RGD-f-PNPs/EPI groups, indicating no obvious toxicity to spleen and no severe immune cell cytotoxicity in all groups. EPI at 6 mg kg$^{-1}$ caused significant body weight loss and histological changes in kidney, liver, and heart, especially. In the hearts, cardiomyocytes demonstrated significant swelling with associated, degenerative, and necrotic areas. Compared with the EPI group, RGD-f-PNPs/EPI-treated animals showed essentially normal kidney and liver morphology, and fewer and smaller necrotic areas in heart, indicating that RGD-f-PNPs/EPI caused no or much less damage in these organs. The histological data suggested that EPI conjugated with RGD-f-PNPs have less cardiotoxicity. Tumor growth in mice treated with RGD-f-PNPs appeared to be similar to that of control group and both EPI and RGD-f-PNPs/EPI groups significantly reduced tumor growth than control group ($p < 0.05$, $p < 0.001$, respectively Fig. 5b). These data showed an enhanced anti-tumor efficacy in vivo, which is likely a result of the specific tumor targeting and efficient delivery of RGD-f-PNPs/EPI.

For any nanoparticle-based therapy, the foremost important issue is not to introduce additional side effects from the nanoparticles themselves. Without EPI loading, the RGD-f-PNP itself indicated good cytocompatibility based on the MTT test (Fig. 4c). The studies also showed that the nanoparticles had no observable immunoresponse (unpublished observation from C57BL/6J mice) and caused no obvious damage in various tissues (Supplementary Fig. 8). In contrast to the EPI treatment group which exhibited significant body weight loss, the mice treated with RGD-f-PNPs/EPI maintained their weight (Fig. 5a). It is well documented that EPI may cause dose-dependent damage to the heart and kidneys. Some reports have linked a high risk of heart failure and myocardial damage to EPI administration even years after stopping the drug[52]. The histologic data on significant damage in kidney and heart tissues confirmed the toxicity of EPI (Supplementary Fig. 8). In this study, RGD-f-PNPs were applied as a nanocarrier to load and deliver EPI with enhanced biodistribution of the drug molecules to have low amounts of drugs in normal tissues and increased amounts in tumors, and enhanced pharmacodynamics (controllable release of the drug at the site of action in the tumor) to produce enhanced efficacy, while simultaneously reducing side effects. Thus, this nanoparticle platform could revive the application of EPI as a powerful chemotherapeutic agent by reducing its cardiovascular and renal toxicities. Since the conjugation interactions between RGD-f-PNPs and EPI are not limited to EPI, similarly, the RGD-f-PNPs/EPI can be employed in principle for loading and tracking the most aromatic drugs for enhanced anti-tumor efficiency. In cultured EC cells, EPI was observed to be released from RGD-f-PNPs within a few hours and reached high intracellular concentrations (Fig. 4d) leaving relatively low drug levels detectable in the extracellular space, which were accompanied by better efficacy of EPI at lower doses (Fig. 4c). In vivo studies further confirmed that RGD-f-PNPs/EPI could achieve similar anti-tumor effects at a lower dose with much less side effects (Fig. 5a, b). It is noteworthy that the dose of EPI at 6 mg kg$^{-1}$ used in this study for mouse is equivalent to mid-low dose used in

patients, which can cause significant cardiotoxicities. While EPI is not often used in ESCC, this study with KYSE-30 cells clearly suggests that EPI could become effective and safe for ESCC patients if it is packed within a suitable nanoparticle drug delivery platform.

**In vivo tumor imaging and tracking of EPI delivery**. To confirm the capability of the RGD-f-PNPs as NIR fluorescence imaging probe in vivo, we intravenously injected RGD-f-PNPs and RGD-f-PNPs/EPI into the tail veins of mice and conducted NIR fluorescent imaging using an IVIS Lumina II imaging system (emission filter: 815–870 nm). After 30 min of RGD-f-PNPs and RGD-f-PNPs/EPI injection, the xenograft tumors were clearly visualized in live animal NIR imaging with relative low background (Fig. 5c). Another significant signal was detected from the abdominal region adjacent to the tumor injection site likely stomach. After 16 h following the RGD-f-PNPs/EPI injection, the fluorescence at the tumor site increased likely due to the release of EPI while the abdominal signal remained unchanged. This phenomenon reflects the in vitro EPI releasing observation (Fig. 4a) due to the nonexistent electrostatic interactions between EPI and RGD-f-PNPs. Later, the abdominal signals were confirmed to be autofluorescence arising from the stomach due to dietary components containing fluorescent ingredients (Fig. 5e). When the mice were fed an Alfalfa-free diet, these autofluorescence signals were reduced (Supplementary Fig. 9). At the end of the experiment, the tumors and five vital organs were collected from all mice followed by ex vivo NIR fluorescence imaging (Fig. 5d). Clear NIR fluorescence was observed in the tumors isolated from animals injected with RGD-f-PNPs and RGD-f-PNPs/EPI, but not in the control or EPI-treated groups (Fig. 5d). It is worthwhile to note that the tumors isolated from the RGD-f-PNPs/EPI group were much smaller than those from the control and RGD-f-PNP-only groups. These data demonstrated that the RGD-f-PNPs/EPI could reach, penetrate, and remain in the tumor sites, and the NIR fluorescence from these peptide nanoparticles was bright enough to be visualized using a standard NIR small animal imaging platform.

## Discussion

Both in vitro and in vivo studies demonstrated that the RGD moieties could facilitate selective targeting of RGD-f-PNPs to EC cancer cells and tumor tissues (Figs. 3 and 5c). More importantly, no definitive signal of RGD-f-PNPs intracellular penetration was observed in cultured non-cancer cells or non-neoplastic tissues in animals. Due to poorly preserved tissue architecture of the isolated tumor samples following processing, identification of intact nanoparticle signal within different tumor compartments (e.g., intracellular vs. interstitial) was difficult to confirm using conventional immunohistochemistry. With fluorescence emission of longer wavelength and development of super-resolution in vivo imaging, it may be possible to image the precise localization and even the subcellular localization of the RGD-f-PNPs for monitoring drug delivery in live animals. Further studies are required to understand how RGD-f-PNPs cross blood vessels and penetrate tumors, how they further uniformly or heterogeneously distribute throughout the tumor, and how this affects further drug release within the tumor, which could help us to maximize nanoparticle-mediated chemo-drug efficacy.

In summary, the RGD-modified peptide nanoparticle system that can generate both visible and NIR fluorescence and directly target EC cells for chemotherapy delivery has been developed. In contrast to other drug delivery systems in the literature[36–39], the proposed drug-embedded nanoparticles, self-assembled by peptides are made of natural amino acids, which have inherent

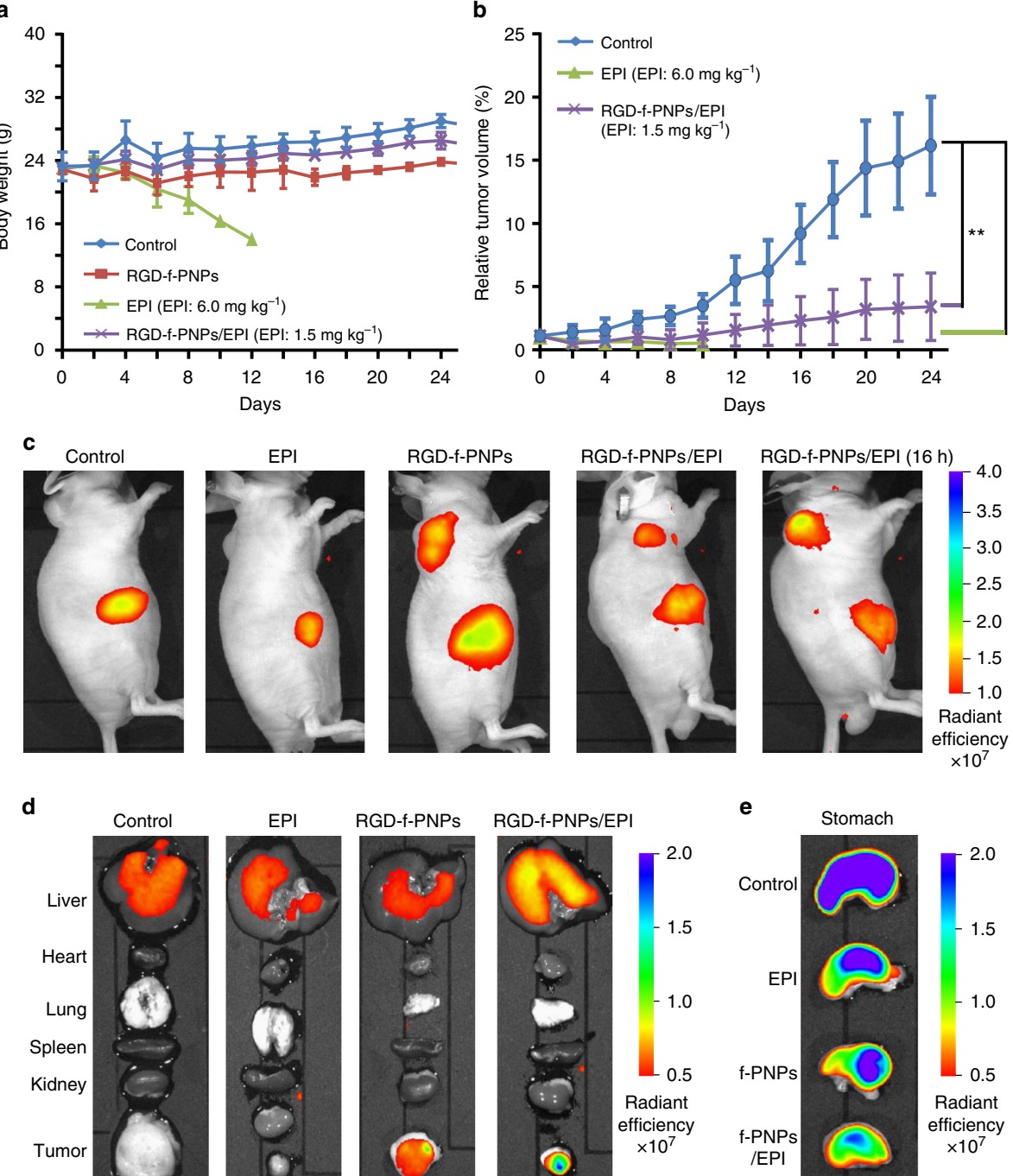

**Fig. 5** In vivo targeted tumor imaging and enhanced anti-tumor efficacy of RGD-f-PNPs/EPI. Tumor-bearing nude mice were treated with various drug formations at the dose of 1.5 mg kg$^{-1}$ (RGD-f-PNPs/EPI) and 6 mg kg$^{-1}$ (EPI) for 3 weeks. **a** Body weight of mice with the various treatments. No obvious side effects for the RGD-f-PNPs were observed in terms of signs of distress, lethality, or drop of body weights, while treatment of EPI resulted in significant lost in body weight and three out of five animals died after 1 week of treatment. Error bars represent s.d. ($n = 5$). **b** Growth curves of tumors in the mice after the various treatments. Tumor growth in mice treated with RGD-f-PNPs appeared to be similar to that of control group and both EPI and RGD-f-PNPs/EPI groups significantly reduced tumor growth than control group ($p < 0.05$). The data indicate the enhanced anti-tumor efficacy of RGD-f-PNPs/EPI in vivo, as a result of its specific tumor targeting and efficient delivery and release process using RGD-f-PNPs. Error bars represent s.d. ($n = 5$). Statistical significance was determined using an unpaired $t$ test in tumor growth curve analysis, with *$p < 0.05$, ** $p < 0.001$. **c** In vivo whole animal imaging of NIR fluorescence after intravenous injection via the tail vein in the form of saline, EPI alone, RGD-f-PNPs alone, and RGD-f-PNPs/EPI. After 30 min of RGD-f-PNPs and RGD-f-PNPs/EPI injection, the xenograft tumors were clearly visualized with NIR imaging at relatively low background. Another significant signal was detected from the abdominal region adjacent to the tumor injection site due to dietary components in the stomach (**e**). After 16 h of the RGD-f-PNPs/EPI injection, the fluorescence at the tumor site increased likely due to the release of EPI while the abdominal signal remained unchanged. **d** Ex vivo NIR fluorescence imaging in tumor and five different organs collected after sacrificing the mice. Clear fluorescence signal was observed from tumor of mice with injection of RGD-f-PNPs and RGD-f-PNPs/EPI, but not in the control or EPI-treated groups. **e** Ex vivo NIR fluorescence imaging of stomach collected after sacrificing the mice. The fluorescence from the stomach was due to the mice diet

biocompatibility and may raise less health and environmental concerns. This technique has the potential to augment the antineoplastic drug effect of EPI or possibly other aromatic chemotherapeutic agents and minimize side effects. The RGD-modified peptide nanoparticles have demonstrated capability to EC tumor targeting in vitro and in vivo. Also, in vivo targeted tumor imaging and EPI delivery could be achieved through intravenous injection of the RGD-f-PNPs/EPI. Compared to EPI alone, the RGD-f-PNPs/EPI nanoparticle composites showed improved anti-tumor activity with significantly fewer side effects. The present work demonstrates the promising translational potential of the RGD-modified cyclic peptide nanoparticles, offering real-time tumor imaging and targeted EPI delivery to tumor cells, thereby enhancing therapeutic effects and reducing the side effects of chemotherapy.

## Methods

**Materials.** Cyclo[-(D-Ala-L-Glu-D-Ala-L-Trp)2-] peptides were purchased from GL Biochem (China) and AnaSpec (USA) with 95% purify. EPI hydrochloride, ICG, and zinc nitrate hexahydrate were purchased from Sigma-Aldrich (USA). Dulbecco's modified Eagle's medium and Eagle's minimum essential medium were purchased from Life Technologies (USA). The CCK-8 cell proliferation reagent was purchased from Dojindo Molecular Technologies (USA). Fetal bovine serum (FBS) and penicillin/streptomycin were purchased from Invitrogen (USA). All other reagents and solvents were used without further purification.

**Cell lines.** Human ESCC KYSE-30, KYSE-150, KYSE-270, and KYSE-410 cell lines were obtained from Dr. C.S. Yang at Rutgers University and they are cultured in RPMI-1640/Ham's F12 (1:1) medium with 5% FBS[53]; EAC cell line OE33 (Sigma-Aldrich, USA) was cultured in RPMI-1640 medium containing 10% FBS; normal esophageal epithelial HET-1A cells (ATCC, USA) were maintained in serum-free LHC-9 medium[53]. ATAC-seq analysis has been conducted in HET-1A and ESCC cell lines. The transcriptome profiling is consistent to published data. Cells are periodically examined to ensure free from mycoplasma contamination using Commercial Detection Kit (Lonza, Switzerland, LT07-703). One week before experiment, HET-1A cells were adapted in experimental culture medium same as that used for ESCC cells. All culture medium contained 1% penicillin/streptomycin and cells were maintained at 37 °C in a humidified 5% CO2 incubator.

**Live cell imaging for cellular uptake of f-PNPs.** Cells were seeded at density of $4 \times 10^5$ cells into glass-bottom dish and maintained in experimental medium for 24 h before imaging. After removal of the medium, cells were incubated within BSS-Ca²⁺ (in mM: 140 NaCl, 2.8 KCl, 2 MgCl2, 10 HEPES, 2 Ca²⁺, pH 7.2) solution containing f-PNPs, EPI, or f-PNPs-EPI, respectively. Zeiss LSM780 confocal fluorescence microscope (Zeiss, Germany) with ×63 water immersion objective (NA 1.2) was used to visualize the fluorescent signals. The excitation/emission wavelengths set for f-PNPs and EPI were 405/430 and 488/561 nm, respectively. All experiments were performed at room temperature.

**CD51 gene knockout.** Two gRNAs (gRNA1, GATTCAATTGGCTGGCACCGG and gRNA2, GGTGACTGGTCTTCTACCCGC) targeting CD51 were cloned into pX330 plasmids[54] (Addgene, #42230). The 1:1 mixed plasmids were transfected into KYSE-30 and OE33 cells using Lipofectamine 3000 (Thermo Fisher Scientific, USA). The knockout efficiency was verified by immunofluorescence staining. .

**Confocal imaging for protein expression and localization.** To verify the expression level of CD51, cells were seeded into glass-bottom dish with thickness of 0.16–0.19 mm (In vitro Scientific, USA). After entering exponential growth curve, the cells were fixed with 4% paraformaldehyde at room temperature for 15 min. After washing with PBS three times, cells were blocked in the 0.1% PBST supplemented with 10% horse serum for 30 min at room temperature. Then, the cells were incubated with mouse anti-CD51 primary antibody (Ancell, USA, 202–820, dilution ratio: 1:100) in blocking solution containing 4% bovine serum albumin and kept overnight at 4 °C. Alexa Fluor® 488-labeled secondary antibodies (Abcam, USA, ab150113, dilution ratio 1:500) were employed to visualize the expression and localization of CD51 protein under Zeiss LSM 780 confocal microscope with ×63 water immersion objective (NA 1.2). Additional images to visualize the expression and localization of CD51 protein of KYSE-30 and OE33 cells (both wild-type and knockout cells) were obtained using Leica DMi8 inverted microscope (Leica, Germany) with GFP filter (Ex: 450–490 nm; Em: 500–550 nm) and ×63 oil immersion objective (NA 1.4).

**Flow cytometry.** Cells were seeded into T25 flask and cultured for 24 h before experiment. After trypsinization, the cells were collected in the PBS and fixed with 4% paraformaldehyde. By washing with PBS, the cells were incubated in the PBS containing 1% bovine serum albumin (BSA) for 30 min at room temperature. Then, the cells were incubated with anti-CD51 antibody (Ancell, USA, 202–820, dilution ratio 1:100) for 1 h at room temperature and labeled with Alexa-488 with secondary antibody (Abcam, USA, ab150113, dilution ratio 1:500) for 1 h in dark. The fluorescence intensity was detected using flow cytometer (Beckman Coulter, USA). For each sample, at least 10,000 cells were collected to analyze the relative fluorescence data of each sample.

**Cytotoxicity assay.** KYSE-30 cells were seeded into 96-well plates at a density of $2 \times 10^3$ per well and maintained for 24 h in the culture medium. Cells were treated with f-PNP, EPI, and f-PNP-EPI at indicated concentrations for another 24 h. Then, 20 μL of 5 mg mL⁻¹ MTT (Sigma-Aldrich, USA) was added into each well and incubated for another 4 h. The medium was removed completely and 200 μL dimethyl sulfoxide was added into each well. The cytotoxicity was evaluated by absorbance at 550 nm by FlexStation 3 (Molecular Device, USA). Each experiment was repeated at least three times.

**Nanoparticle self-assembly.** Fresh stock solutions of the cyclic peptides were prepared by dissolving the lyophilized from the peptides in isopropanol at a concentration of 10 mg mL⁻¹. Fresh stock solutions were prepared for each experiment to avoid pre-aggregation. Cyclic peptide/Zn²⁺ nanoparticles were obtained from the reaction of equivalent amount of cyclic peptides and ZnCl2 dissolved in a mixture of 80% 0.01 M aqueous NaOH and 20% methanol. The reactants were heated to 85 °C for 30 min at a rate of 2 °C min⁻¹ and cooling at a rate of 1 °C min⁻¹ using sealed autoclave. Filtration and centrifugation were applied later to purify the self-assembled f-PNPs.

**Atomic force microscopy.** The nanomorphology of the PNPs was characterized using an MFP-3D AFM system with an ACTA-50 Probe (AppNano, Mountain View, CA, USA). The silicon tip is pyramidal shaped with a radius of curvature below 10 nm. The silicon cantilever is rectangular shaped with alumina reflex coating. The spring constant of the cantilever is 13–77 N m⁻¹, and the frequency is 200–400 kHz. Images were recorded using the Igor Pro software in AC mode at room temperature, which minimized the distortion due to the mechanical interactions between the AFM tip and the surface.

**Scanning electron microscopy.** A 10 μL droplet containing 1 mg mL⁻¹ peptide nanoparticles was dried at room temperature on a microscope silicon chip. Samples were observed using FEI Nova Nano SEM 400 scanning electron microscopy (FEI, Hillsboro, OR, USA) at 20 kV.

**Dynamic lighting scattering.** A Zetasizer Nano ZS (Malvern Instruments, Malvern, UK) was used for all dynamic lighting scattering (DLS) measurements. In order to determine nanoparticle size distribution, 1 mL sample was placed in a glass cuvette and DLS was performed at 25 °C with a backscatter angle of 173° and an equilibration time of 120 s. The intensity-based z-averaged hydrodynamic diameters were reported based on five scans. The measurement was conducted at 25 °C and five cycles were performed to improve accuracy.

**Fourier-transformed infrared spectroscopy.** Spectra were recorded on a Perkin-Elmer Spectrum Spotlight 300R microscopy (Perkin-Elmer, Waltham, MA, USA). The samples were spread and vacuum dried on a polyethylene film. Spectra were corrected for absorption from a phosphate buffer blank sample. Measurements were taken using 3 cm⁻¹ resolution and by averaging 38 scans.

**Powder X-ray diffraction.** Spectrum was recorded with Bruker D8 Advance X-ray powder diffractometer (Bruker, Billerica, MA, USA) at room temperature, scan range 5–45° 2θ, count 2 s. PNP powder was placed on the standard flat sample refection holder. MDI Jade software controls data collection and allows data analysis.

**Fluorescence spectroscopy.** Fluorescence excitation and emission spectra were measured on a Cary Eclipse Fluorescence Spectrophotometer (Agilent Technologies, Santa Clara, CA, USA) with light measured orthogonally to the excitation light. The excitation and emission slit widths were set to 5 and 5 nm.

**Preparation of EPI-loaded RGD-f-PNPs.** Conjugation of EPI to the RGD-f-PNPs was achieved through π–π stacking. The aromatic group within EPI would tend to stack with the aromatic amino acids from the RGD-f-PNPs. Specifically, 100 μL of 2.5 mg mL⁻¹ EPI solution was added to the 1 mL of 1 mg mL⁻¹ RGD-f-PNPs suspensions and kept overnight at room temperature to complete the reaction. Fluorescence emission and absorbance spectra of EPI were recorded before and after RGD-f-PNP conjugation. The reaction solution was then centrifuged at 16,873 × g for 20 min and the precipitates were washed three times with distilled water to remove any excess EPI and salts. The precipitated RGD-f-PNPs/EPI conjugates were re-dissolved in PBS. The EPI releasing was monitored through absorbance spectra and fluorescence microscopy.

**Animal xenograft model establishment and tumor assay**. Animals care and experiments were approved by the Institutional Animal Care and Utilization Committee (IACUC). In brief, 50 µL of $1 \times 10^6$ KYSE-30 cells in PBS were mixed with equal volumes of Matrigel (Corning, MA, USA), and then subcutaneously injected into the back of each 6-week-old male NCr nu/nu nude mouse (Taconic Farm, NY, USA). A total of 20 mice were randomly assigned into four groups with each group containing five animals. One week after the cancer cell injection, the mice were treated via intraperitoneal injection as follows: control, PBS only; RGD-f-PNPs, nanoparticle only, 6 mg kg$^{-1}$; EPI, EPI 6 mg kg$^{-1}$; RGD-f-PNPs/EPI, nanoparticle combined with EPI including 6 mg kg$^{-1}$ NP and 1.5 mg kg$^{-1}$ EPI. Tumor volume parameters and body weight were collected every other day after initial treatment. The tumor volume (mm$^3$) was calculated by the following formula: volume = (width)$^2$ × length × 3.14/6. At the end of experiment or the tumor size reaching 15 mm in diameter, animals were euthanized, and the tumor, heart, spleen, kidney, and liver were collected. The tumor size variances were similar between the groups. The tumor growth curve for each animal was normalized with its tumor volume at day 0 just before treatment. In another set of in vivo experiment, KYSE-30 cells transfected with plasmids containing anti-CD51 gRNAs were used for injection. To avoid autofluorescence from food, a special Alfalfa-free diet (Envigo, USA) was used to feed the mice for 2–3 days in some of the fluorescent imaging experiments.

**In vivo NIR imaging**. In vivo NIR imaging was performed using an IVIS Spectrum Imaging System (Perkin-Elmer). The fluorescent light emitted from the mice was detected by a CCD camera. The acquisition and analysis of the data were performed using the Living Image 4.2.1 Software. The mice were injected with PBS, EPI, RGD-f-PNPs, and RGD-f-PNPs/EPI through the tail vein. Fluorescence signals were recorded with the mice lightly anesthetized (2.5% isoflurane in oxygen flow, 1.5 L min$^{-1}$).

**Hematoxylin and eosin staining**. Once removed from mice, the tissues were immediately fixed in 10% neutral-buffered formalin for 48 h in 4 °C following standard H&E protocol. After paraffin embedding, dewaxing, and hydration, the tissue slides were processed as follows: hematoxylin solution staining 3 min, DD water wash 1 min, incubated in 1% HCL-ethanol 30 s, DD water wash 1 min, 1% ammonia solution 10 s, DD water wash 1 min, eosin solution staining 10 min, DD water wash 1 min, 70% ethanol 30 s, 80% ethanol 1 min, 95% ethanol 1 min, 100% ethanol I 2 min, 100% ethanol II 2 min, xylene I 5 min, xylene II 5 min. After that, the slides were sealed with neutral resin and observed under DMi8 inverted microscope with stitching function (Leica, Germany).

**Statistical analysis**. Results are presented as mean or mean ± standard deviation (s.d.). One-way analysis of variance or $t$ test was applied to evaluate the significance among groups. In all cases, a $p$ value <0.05 is considered to be statistically significant.

**Data availability**. All relative data are available from the corresponding authors upon request.

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

## Acknowledgements

This work was supported by research grants from the National Institutes of Health R01-CA226251 to M.Z., National Institutes of Health R01-CA185055 and University of Texas System STARs Award to Z.P. and the Ohio State University Comprehensive Cancer Center Pelotonia Postdoctoral Fellowship to Z.F. We thank Dr. Shuqian He for her assistance on pathological analysis and Ms. Marina Chu for her editing.

## Author contributions

Z.F., Z.P., and M.Z. conceived and designed the experiments, Z.F., Y.C., C.C., and L.S. performed the experiments, all the co-authors analyzed the data, Z.P. and M.Z. contributed the materials and analysis tools, and Z.F., Y.C., C.C., D.W., Z.P., and M.Z. analyzed the results and wrote and edited the manuscript. All the co-authors discussed and commented on the manuscript.

## Additional information

**Competing interests:** The authors declare no competing interests.

