## [Peer Review File · Nature Communications]

Reviewers' Comments:

Reviewer #1:

Remarks to the Author:

This manuscript by Fan et al reports an exciting delivery system for treating esophageal cancer. RGD-f-PNPs were designed to deliver chemotherapeutic agents to esophageal cancer through the interaction between RGD and $\alpha(v)\beta(3)$ integrin. Epirubicin was tested as an example of such chemotherapeutic agents. Both in vitro and in vivo data suggested specificity and anti-cancer efficacy of the delivery system. Overall this study is quite significant in its potential translational value. However, this reviewer as a biologist can only comment on the biology of this manuscript due to lack of chemical/engineering expertise.

1. Since the cell specificity depends on the recognition of $\alpha(v)\beta(3)$ integrin by RGD, it would be nice to do a simple knockdown experiment on KYSE30 or 270 cells to show that the specificity of RGD-f-PNPs depends on the presence of integrin.
2. Epirubicin is a chemotherapeutic agent commonly used for esophageal adenocarcinoma or gastroesophageal junction adenocarcinoma. Testing this compound for esophageal squamous cell carcinoma cells (KYSE cells) needs to be justified.
3. The doses of 1.5mg/kg RGD-f-PNPs/EPI and 6mg/kg EPI need to be justified. Mouse esophagus may serve as control of toxicity to the normal esophageal epithelial cells. Alfalfa free diet is needed to eliminate autofluorescence in the stomach area.
4. This manuscript contains quite some grammatical and spelling errors. English editing is a must.
5. In Introduction, it is not correct that "currently EC diagnosis is primarily based on individual surgeon's personal experience to interpret patients' signs of symptoms ..." It will surely help to let a clinician with esophageal cancer experience go through this section and highlight the potential translational value.

Reviewer #2:

Remarks to the Author:

This manuscript reports on nanoparticles made of cyclic peptide that, together with RGD and Zinc ions, can generate both fluorescence and directly target EC cells for chemo-drugs delivery and improve antitumor activity. This manuscript is most likely a continuation to the group's former paper: "Bioinspired fluorescent dipeptide nanoparticles for targeted cancer cell imaging and real-time monitoring of drug release". I would suggest to refer the following comments:

- (1) Usually, intrinsic fluorescence is considered fluorescence of the material itself, without any guest material supplement. See for example the paper: "Reconstructive Phase Transition in Ultrashort Peptide Nanostructures and Induced Visible Photoluminescence". In this manuscript, both RGD and Zinc ions generate the fluorescence and NOT the peptide material itself. Thus, I think that the word "intrinsic" is misleading.
- (2) There are several reports on fluorescent peptide-based materials that are relevant to this manuscript, for example "Peptide-Conjugated Fluorescent Silicon Nanoparticles Enabling Simultaneous Tracking and Specific Destruction of Cancer Cells". The authors should explain the difference between the development to this paper.
- (3) The role of the Zinc ions is not clear - if the RGD produces the fluorescence, why do you need the Zinc ions?

To summarize, I recommend to ACCEPT the paper to Nat. Comm., after referring to these comments.

We appreciate both reviewers' recognition regarding the significance and potential translational value of this study. We also thank them for their critical review and constructive comments on the manuscript, which have helped us to design more rigorous experiments and reshape the manuscript. The current version has incorporated some new data on esophageal adenocarcinoma cells, *in vitro* and *in vivo* studies of esophageal cancer cell lines with knockout of integrin alpha-V gene, and toxicity analysis of mice esophageal epithelial. The detailed point-by-point responses are enclosed below. With these new data and substantial changes in the manuscript according to the reviewers' suggestions, we hope this revised version of our manuscript can be published in Nature Communication.

Responses to Reviewer #1:

“This manuscript by Fan et al report an exciting delivery system for treating esophageal cancer. RGD-f-PNPs were designed to deliver chemotherapeutic agents to esophageal cancer through the interaction between RGD and alpha(v)beta(3) integrin. Epirubicin was tested as an example of such chemotherapeutic agents. Both *in vitro* and *in vivo* data suggested specificity and anti-cancer efficacy of the delivery system. Overall this study is quite significant in its potential translational value. However, this reviewer as a biologist can only comment on the biology of this manuscript due to lack of chemical/engineering expertise.”

Respond: We appreciate your encouraging comments.

1. Since the cell specificity depends on the recognition of alpha(v)beta(3) integrin by RGD, it would be nice to do a simple knockdown experiment on KYSE30 or 270 cells to show that the specificity of RGD-f-PNPs depends on the presence of integrin.

Respond: Thank you for the suggestion. We have performed CRISPA/Cas9 based gene knockdown assay and the new data are presented in new **Supplementary Fig. 7**. The effect of knockout of alpha(v)beta(3) integrin (also called CD51) was confirmed by immunofluorescent microscopy imaging. The cells lacking CD51 showed no plasma membrane fluorescent signal from the RGD-f-PNPs. But since CD51 is essential for tumor formation and KYSE-30 cells lacking CD51 couldn't form tumor, we were not able to compare the RGD-f-PNP uptake in xenograft mice injected with KYSE-30/CD KO cells.

2. Epirubicin is a chemotherapeutic agent commonly used for esophageal adenocarcinoma or gastroesophageal junction adenocarcinoma. Testing this compound for esophageal squamous cell carcinoma cells (KYSE cells) needs to be justified.

Respond: We agree. Epirubicin can effectively induce cell death in many cancer cells including esophageal squamous cell carcinoma. Although it has been used often for patients with esophageal adenocarcinoma and gastroesophageal junction adenocarcinoma, more clinical trials demonstrated their benefits for ESCC patients as well (*Andreyev et al, Eur J Cancer. 1995; Honda et al, Dis Esophagus. 2010; Kosugi, et al Scand J Gastroenterol. 2005*). We included a set of data on esophageal adenocarcinoma cell line OE33. As shown in **Supplementary Fig. 7**, the esophageal adenocarcinoma cells (OE33) have higher expression of alpha(v)beta(3) integrin, and RGD-f-PNPs preferentially bind to these cells. Taking together, our data suggest that the RGD-f-PNPs/EPI is suitable for both esophageal squamous cell carcinoma and adenocarcinoma. Thus, this study provides evidence to show that EPI embedded RGD-f-PNPs could be a chemotherapeutic agent for ESCC. We included these points into this revised manuscript.

3. The doses of 1.5mg/kg RGD-f-PNPs/EPI and 6mg/kg EPI need to be justified. Mouse esophagus may serve as control of toxicity to the normal esophageal epithelial cells. Alfalfa free diet is needed to eliminate autofluorescence in the stomach area.

Respond: Clinically, EPI is used 10-120 mg/m², or 0.27-3.24 mg/kg body weight. We selected a mid-low dose (0.5 mg/kg) and converting these data into mouse dose ~6 mg/kg (0.5 multiplied by factor of 12.3) (*Nair AB, et al, 2016, J Basic Clin Pharm 7(2): 27-31*). Since our pilot study showed that 1.5mg/kg RGD-f-PNPs/EPI could achieve similar anti-tumor effect to 6mg/kg EPI, thus this dose was used throughout the *in vivo* study. The enhanced therapeutic effects are due to the enriched aggregation of RGD-f-PNPs/EPI in tumor tissues *via* enhanced permeation and retention (EPR) effect and the high affinity of modified RGD moieties to overexpressed integrin $\alpha_v\beta_3$ on the tumor cells. Moreover, the drug delivery to tumor sites and therapeutic responses could be monitored with NIR fluorescence using RGD-f-PNPs/EPI. We have performed H&E staining in isolated esophageal epithelium removed from control, EPI and RGD-f-PNPs/EPI treated mice. Data showed that there was no obvious toxicity to epithelium from all groups (**Supplementary Fig. 8**). The autofluorescence in stomach has been dramatically reduced in animals fed with Alfalfa free diet (**Supplementary Fig. 10**).

4. This manuscript contains quite some grammatical and spelling errors. English editing is a must.

Respond: Thanks! We have carefully edited the manuscript with professional English editing to eliminate the grammatical and spelling errors.

5. In Introduction, it is not correct that "currently EC diagnosis is primarily based on individual surgeon's personal experience to interpret patients' signs of symptoms ..." It will surely help to let a clinician with esophageal cancer experience go through this section and highlight the potential translational value.

Respond: Thank you for the suggestion. Dr. David Wang, a clinician scientist in University of Texas Southwestern Medical Center with expertise in esophageal cancer has joined this group and provided clinical and translational input. Please see the revised Introduction with all the changes.

Responses to Reviewer #2:

Reviewer #2 (Remarks to the Author):

This manuscript reports on nanoparticles made of cyclic peptide that, together with RGD and Zinc ions, can generate both fluorescence and directly target EC cells for chemo-drugs delivery and improve antitumor activity. This manuscript is most likely a continuation to the group's former paper: "Bioinspired fluorescent dipeptide nanoparticles for targeted cancer cell imaging and real-time monitoring of drug release".

Response: Thank you for the comments. Yes, you are right. We have revised the manuscript according to your comments.

I would suggest to refer the following comments:

(1) Usually, intrinsic fluorescence is considered fluorescence of the material itself, without any guest material supplement. See for example the paper: "Reconstructive Phase Transition in Ultrashort Peptide Nanostructures and Induced Visible Photoluminescence". In this manuscript, both RGD and Zinc ions generates the fluorescence and NOT the peptide material itself. Thus, I think that the word "intrinsic" is misleading.

Response: Thank you for your suggestion. In this revision, the fluorescence of RGD-f-PNPs was not generated by the peptide itself. We agree with your suggestion and have removed the terminology "intrinsic" in the revision.

(2) There are several reports on fluorescent peptide-based materials that are relevant to this manuscript, for example "Peptide-Conjugated Fluorescent Silicon Nanoparticles Enabling Simultaneous Tracking and

Specific Destruction of Cancer Cells". The authors should explain the difference between the development to this paper.

Response: Thank you for your suggestion. We have cited and explained the difference between the related papers and the revised manuscript. In most reports on fluorescent peptide-based materials, peptides are usually utilized as functional agents for their biological activities. Meanwhile, the RGD-f-PNPs developed in our group possess fluorescence property themselves and do not require additional modification with QDs or fluorophores.

(3) The role of the Zinc ions is not clear - if the RGD produce the fluorescence, why do you need the Zinc ions?

Response: Thank you for your comment. Zinc-coordination was applied to stabilize the zinc chelation structure and limit further energy dissipation during thermal relaxation pathways for better quantum yield and fluorescence intensity (*Barondeau DP, et al, 2002, J. Am. Chem. Soc. 124(14):3522-3524*).

To summarize, I recommend to ACCEPT the paper to Nat. Comm., after referring to these comments.

Response: Thank you very much!

Reviewers' Comments:

Reviewer #1:

Remarks to the Author:

All comments have been addressed adequately.